# Flavonoids as Human Intestinal α-Glucosidase Inhibitors

**DOI:** 10.3390/foods10081939

**Published:** 2021-08-20

**Authors:** Elizabeth Barber, Michael J. Houghton, Gary Williamson

**Affiliations:** Department of Nutrition, Dietetics and Food, Faculty of Medicine, Nursing and Health Sciences, School of Clinical Sciences, Monash University, BASE (Be Active Sleep Eat) Facility, 264 Ferntree Gully Road, Notting Hill, VIC 3168, Australia; elizabeth.barber@monash.edu (E.B.); michael.houghton@monash.edu (M.J.H.)

**Keywords:** maltase, sucrase, isomaltase, HPAE-PAD, polyphenols, quercetin, quercetagetin, kaempferol, galangin, acarbose

## Abstract

Certain flavonoids can influence glucose metabolism by inhibiting enzymes involved in carbohydrate digestion and suppressing intestinal glucose absorption. In this study, four structurally-related flavonols (quercetin, kaempferol, quercetagetin and galangin) were evaluated individually for their ability to inhibit human α-glucosidases (sucrase, maltase and isomaltase), and were compared with the antidiabetic drug acarbose and the flavan-3-ol(−)-epigallocatechin-3-gallate (EGCG). Cell-free extracts from human intestinal Caco-2/TC7 cells were used as the enzyme source and products were quantified chromatographically with high accuracy, precision and sensitivity. Acarbose inhibited sucrase, maltase and isomaltase with IC_50_ values of 1.65, 13.9 and 39.1 µM, respectively. A similar inhibition pattern, but with comparatively higher values, was observed with EGCG. Of the flavonols, quercetagetin was the strongest inhibitor of α-glucosidases, with inhibition constants approaching those of acarbose, followed by galangin and kaempferol, while the weakest were quercetin and EGCG. The varied inhibitory effects of flavonols against human α-glucosidases depend on their structures, the enzyme source and substrates employed. The flavonols were more effective than EGCG, but less so than acarbose, and so may be useful in regulating sugar digestion and postprandial glycaemia without the side effects associated with acarbose treatment.

## 1. Introduction

One of the earliest signs of type 2 diabetes (T2D) is elevated and erratic postprandial glycaemia that promotes oxidative stress at various sites within the body [1]. Controlling postprandial glycaemia is an important strategy in the management of T2D. One way is by slowing down carbohydrate digestion and glucose absorption in the intestine via the inhibition of salivary/pancreatic α-amylases and membrane-bound brush-border α-glucosidases.

There are four relevant types of digestive α-glucosidases in humans, maltase (α-1,4-glucosidase; EC 3.2.1.20), glucoamylase (exo-1,4-α-glucosidase; EC 3.2.1.3), sucrase (α-glucohydrolase; EC 3.2.1.48) and isomaltase (oligo-1,6-glucosidase or α-dextrinase; EC 3.2.1.10). Maltase and glucoamylase have a unique, high α-1,4 hydrolytic activity for longer chain maltooligosaccharides to produce glucose [2], and are referred to as maltase/glucoamylase (MGAM) [3]. Sucrase-isomaltase (SI) is synthesized as a single glycoprotein chain in intestinal cells [4], and then cleaved into individual sucrase and isomaltase domains that reassociate non-covalently. Sucrase hydrolyses α-1,2-glycosidic bonds in sucrose to produce glucose and fructose. Isomaltase is the only enzyme able to hydrolyze the α-1,6-glycosidic linkage in α-limit dextrins to produce glucose. MGAM and SI complexes are located along the entire small intestine [5,6] and function to catalyze the production of glucose and fructose from disaccharides, dextrins and dietary polysaccharides. Glucose and fructose pass across intestinal cell membranes via glucose transporters (GLUTs), mainly sodium-glucose transport protein-1 (SGLT1) and glucose transporters -2 and -5 (GLUT2, GLUT5). The pathway of carbohydrate hydrolysis and absorption in the intestine is summarized in Figure 1. Rapidly digested and absorbed glucose in the intestine results in a sharp increase in plasma glucose, which is regulated by insulin-stimulated uptake of glucose into tissues.

The most commonly used FDA-approved pharmaceutical α-glucosidase inhibitor is acarbose, a fermented product from *Actinoplanes* species [7] that is low-risk and non-toxic [8,9], but is associated with uncomfortable side effects such as bloating, cramping, flatulence and abdominal pain [10], and drug-intolerance with chronic treatment [11]. Several potent α-glucosidase inhibitors from plant sources have been identified and have received great attention from the scientific community worldwide as they possess no evident side effects [12,13]. Among them were flavonoids, the most extensively studied compounds as natural antidiabetic agents, associated with a reduction in risk of diabetes in humans, animals and in vitro models [14,15]. Food-derived flavonoids show extremely low toxicity [16,17,18].

Flavonoids are found ubiquitously in plants and represent ~60% of all dietary (poly)phenolic compounds [19,20]. Flavonols, a sub-class of flavonoids, are present in onions, kale, apples, berries, leeks and broccoli [19]. Some flavonols excreted in urine can be used as biomarkers of flavonol intake and are significantly associated with a lower T2D risk [21]. Many flavonoids extracted from plants inhibit α-amylase and α-glucosidases activities in vitro and improved postprandial glycaemia in diabetic animal models and limited human studies [22,23]. Very few studies have reported on the inhibition of isomaltase, however. The disaccharide isomaltose is rarely present in nature but is commonly added as low-caloric food sweeteners in industrial-scale production [24,25], or produced from amylopectin hydrolysis to α-limit dextrins. Studies assessing the isomaltase inhibitory potential by flavonoids and acarbose are therefore of interest.

Unfortunately, many enzyme inhibition studies have been conducted using α-glucosidases from yeast or bacteria, with fewer studies using human intestinal enzymes. The inhibition of yeast and human α-glucosidases is very different, specific to the type of substrate, as reported for maltose [2]. Here we used Caco-2 cells, originating from human colon cancer cells, which form monolayers that differentiate to produce apical microvilli with high expression of maltase and sucrase. The Caco-2/TC7 clone specifically expresses high SI levels at 19–25 days post-confluence [24,26]. Using an enzyme preparation from these cells, we have evaluated sucrase, maltase and isomaltase inhibition by several flavonols and compared them to acarbose and (−)-epigallocatechin-3-gallate (EGCG), a flavan-3-ol known for its inhibitory activity on sucrase and maltase of various sources [27]. These natural compounds may provide promising alternatives for diabetes management with no undesirable side effects.

## 2. Materials and Methods

### 2.1. Reagents and Instruments

Buffer components, sugar substrates and standards, and most inhibitors (acarbose, galangin, kaempferol and EGCG) were purchased from Sigma-Aldrich Corp., Merck (St. Louis, MO, USA), with purity >98%. Quercetagetin was purchased from EMD Millipore, Merck (Burlington, MA, USA) and quercetin was purchased from Extrasynthese (Genay, France). Maltose monohydrate, sucrose and isomaltose were used as substrates for the enzyme assay, and together with fructose and glucose, were used as sugar standards for the chromatographic analyses. All other chemicals were of analytical grade and also purchased from Sigma-Aldrich, unless specified otherwise.

The Dionex™ Integrion™ HPIC™ (High Performance Ion Chromatography) system was used for High-Performance Anion-Exchange chromatography with Pulsed Amperometric Detection (HPAE-PAD) (Thermo Fisher Scientific Inc., Waltham, MA, USA) for the separation and analysis of sugars. A PHERAstar FS plate reader (BMG Labtech, Ortenberg, Germany) was used for measuring absorbance in the total protein assay. High-purity (18.2 mΩ/cm) H_2_O supplied by a MilliQ system (Millipore) was used throughout.

### 2.2. HPAE-PAD Instrumentation and Chromatographic Conditions

Disaccharides and monosaccharides were analyzed by HPAE-PAD on the Dionex™ Integrion™ HPIC™ system (Thermo Fisher Scientific). Separation of the carbohydrates was achieved on a CarboPac PA210 column (2 × 150 mm), preceded by a CarboPac PA210 Guard column (2 × 30 mm), with the column and compartment temperatures maintained at 30 °C and 20 °C, respectively. Eluent was generated using a Dionex™ EGC 500 KOH eluent generator cartridge with Dionex™ continuously regenerated-anion trap column 600, with eluent concentration following a multistep gradient: 12 mM for 12 min, 100 mM for 8 min and 12 mM for 12 min, at a flow rate of 0.2 mL/min and with a sample injection volume of 2.5 μL. Detection was performed using a gold working electrode and AgCl reference electrode at pH ~ 12.0, with a collection rate of 2.00 Hz using the “Gold, Carbo, Quad” waveform. The total run time per sample was 32 min, with 3 min allowed between analyses for sample injection by autosampler. A wash injection of only H_2_O while following the same multistep eluent gradient was performed at the end of every batch (12–14 samples). All injections were performed in duplicate and peak identification was achieved by comparing retention times to the standards. Dionex™ Chromeleon™ 7 Chromatography Data System, version 7.2.9 (Thermo Fisher Scientific, Waltham, MA, USA), was used to process the chromatograms, ensuring peaks were suitably integrated before recording peak area. Sugars were quantified from peak areas using standard curves, with the standards prepared in the same buffer as the samples.

### 2.3. Substrate, Inhibitor/Flavonoid and Sample Preparation

Stock solutions (250 mM) of sucrose, maltose and isomaltose (substrates), plus glucose and fructose (standards only), were prepared in sodium phosphate buffer (SPB, 10 mM, pH 7.0). Stock solutions of acarbose (1 mM) and all flavonoids (10 mM) were prepared by dissolving in their respective solvents, stored at −20°C and used within 2 weeks. Acarbose and EGCG were prepared in H_2_O, quercetin and quercetagetin were prepared in DMSO, while kaempferol and galangin were dissolved in absolute ethanol. The maximum working concentrations for each compound was pre-determined before enzymatic reaction assay to ensure no precipitation occurred in the system. Compared to acarbose and EGCG, lower solubility of tested flavonols was expected due to their structural differences, where compounds with less than 100 µg/mL solubility were considered poorly soluble, as reported previously [28,29]. Inhibitors were tested at various concentrations: acarbose (0.1–100 µM), EGCG (5–1500 µM), quercetin (20–200 µM), quercetagetin (1–50 µM), kaempferol (5–40 µM) and galangin (1–25 µM). The flavonols were tested up to their maximum soluble concentrations. Working solutions were prepared fresh at various concentrations in SPB buffer immediately before assaying. The maximum concentrations (*v*/*v*) of DMSO were ≤2% and ≤0.5% for quercetin and quercetagetin, respectively, and ethanol was ≤0.5% for kaempferol and galangin. The solvents did not affect enzyme activity, as demonstrated by vehicle controls. Cell-free extracts (CFE) were prepared as described in Section 2.6 and used as the enzyme source.

All prepared standards and assay samples, for both method validation and post-assay quantification, underwent the same treatment prior to injection on the HPAE-PAD system. All were deproteinated by mixing with an equal volume of acetonitrile, vortexed for 30 s and centrifuged at 17,000× *g,* 15 min at 4 °C. The resulting supernatants were then diluted at least 10× in H_2_O (maximum final acetonitrile concentration of 5% (*v*/*v*)). Additionally, all standards and samples containing enzyme and substrates were filtered through 0.2 µM polyether sulfone (PES) filters (Sartorius, Göttingen, Germany). All standards, samples and blanks were kept at 4–8 °C until analysis by HPAE-PAD, as described in Section 2.2, was complete.

### 2.4. Validation Parameters for Quantification

The HPAE-PAD method for the quantification of glucose, sucrose, fructose, isomaltose and maltose was set up based on our previously published method [30], but with improved sensitivity and run time. The method was validated for specificity, linearity, sensitivity, precision, and accuracy as percent extraction recovery, according to the guidelines issued by the U.S. Food and Drug Administration (FDA) 2018 [31] and International Conference on Harmonisation (ICH) 2005 [32].

#### 2.4.1. Specificity and Matrix Effect

Specificity was determined by evaluating any endogenous interferences from the CFE. A comparison study was conducted on chromatograms of a blank incubated CFE matrix sample (CFE/enzyme only), CFE incubation sample with inhibitors (without substrate), CFE incubation sample with a substrate (without inhibitor), individual inhibitors only and blank assay solvent (DMSO and ethanol (*v*/*v* <2%)). Blank samples spiked with a known amount of maltose, sucrose and isomaltose served as reference. The matrix effect was evaluated by comparing the analytical response of sugar spikes in SPB to those in H_2_O, to ensure accurate calibration plots were constructed.

#### 2.4.2. Linearity

The linearity of the HPAE-PAD method was evaluated by a calibration curve constructed by plotting concentrations of standards against their peak areas within the determined limits of detection and quantification (LOD and LOQ, respectively; see below). Six different concentrations of maltose, sucrose and isomaltose (0.1, 0.5, 1.0, 2.5, 5.0 and 10.0 µg/mL) were prepared in assay incubation buffer or distilled water and measured in triplicate on four different days, giving a total of twelve replicates to construct the curve. Linearity was evaluated by calculating a regression line by the least-squares method, determining a linear equation (Equation (1)), where y = peak area, x = concentration, a = slope, b = intercept, and R^2^ for each standard [33].
(1)y=ax+b

#### 2.4.3. Sensitivity

The sensitivity of the method was evaluated by determining the LOD and LOQ using data generated from the calibration curve. LOD and LOQ were measured using the SD of the y-intercept and the slope of the calibration curve, as shown in Equations (2) and (3) below, where ${\mathrm{SD}}_{y-\mathrm{int}}$ is the standard deviation of the y-intercept and S is the calibration curve slope. Both LOD and LOQ were expressed as analyte concentration (µM):(2)$$\mathrm{LOD}=3.3\times \frac{{\mathrm{SD}}_{y-\mathrm{int}}}{S}$$
(3)$$\mathrm{LOQ}=10\times \frac{{\mathrm{SD}}_{y-\mathrm{int}}}{S}$$

#### 2.4.4. Precision

Precision (repeatability and reproducibility) was determined through the analysis of intra- and inter-day assay using standards in SPB quantification buffer. Intra-assay precision was assessed by measuring six concentrations of each standard measured in triplicate on the same day, in one laboratory by one person. Inter-assay precision was carried out by measuring the same concentrations of standards measured (in triplicate) over four different days by two analysts in the same laboratory. Precision was expressed as percent coefficient of variance (%CV), according to Equation (4), calculated as:(4)$$\%\mathrm{CV}\;=\frac{\mathrm{standard}\;\mathrm{deviation}}{\mathrm{sample}\;\mathrm{mean}}\times 100$$

#### 2.4.5. Accuracy as Extraction Recovery

As standard reference material was not used, accuracy was determined using the extraction recovery calculated by comparing the analytical response of two different concentrations of substrates spiked pre-assay to the values recovered post-assay in triplicates. The accuracy was calculated following Equation (5) as below:(5)$$\mathrm{Extraction}\;\mathrm{Recovery}\;\left(\%\right)=\frac{C_{\mathrm{measured}}}{C_{\mathrm{standard}}}\times 100$$
where $C_{\mathrm{measured}}$ = measured concentration calculated from the calibration curve (µM); $C_{\mathrm{standard}}$ = real (prepared) concentration of the standard solution (µM), which was used in the calculation of the percent relative error (%RE), as shown in Equation (6) below:(6)$${\mathrm{Relative}\;\mathrm{Error}}_{\mathrm{accuracy}}\;\left(\%\mathrm{RE}\right)=\frac{C_{\mathrm{measured}}-{\;C}_{\mathrm{standard}}}{C_{\mathrm{standard}}}$$

### 2.5. Cell Culture

Caco-2/TC7 cells, originating from human colon adenocarcinoma, were a kind donation from Dr Rousset, INSERM U505, Paris, France. Cells were seeded at ~1 × 10^6^ cells/T75 culture flask, maintained at 37 °C in an atmosphere of 10% CO_2_/90% air at a relative humidity in Dulbecco’s modified Eagle’s medium (DMEM) with high glucose (4.5 g/L) (Sigma-Aldrich, St. Louis, MO, USA). The medium was supplemented with 2% (*v*/*v*) Glutamax (Invitrogen, Thermo Fisher Scientific, Waltham, MA, USA), 1% (*v*/*v*) nonessential amino acids, 1% (*v*/*v*) penicillin-streptomycin (100 IU/mL penicillin and 100 ug/mL streptomycin) and 20% (*v*/*v*) heat-inactivated foetal bovine serum (FBS; 56 °C, 30 min) (Gibco, Thermo Fisher Scientific, Waltham, MA, USA). Cells were passaged with 0.25% trypsin-EDTA before reaching ~70% confluence. Cells were used for experiments during passages 31–34 and were maintained until 21 days post-confluence, with medium routinely changed every 2–3 days, to differentiate. On day 21, cells were thoroughly washed twice with ice-cold PBS and harvested into ice-cold PBS containing 1% protease inhibitor cocktail (Sigma-Aldrich, St. Louis, MO, USA). Following centrifugation at 200× *g*, 10 min at 4 °C, the supernatant was discarded and cell pellets were frozen immediately at −80 °C until required for the enzyme assay.

### 2.6. Enzyme Activity Assay

To mimic intestinal digestion, an in vitro assay using Caco-2/TC7 cell extracts containing sucrase, maltase and isomaltase was conducted. Frozen cells were thawed, 1 mL ice-cold SPB added and then passed through a 21-G needle 15–20 times. The lysate was centrifuged at 14,000× *g*, 10 min at 4 °C and the supernatant containing cell-free extract (CFE) collected. Total protein concentration in the CFEs was determined by Bradford assay [34,35], using the Pierce Coomassie Bradford reagent and BSA standards (Thermo Fisher Scientific, Waltham, MA, USA).

Assay mixtures, total volume 250 µL, containing CFE (final protein concentration at 0.1–0.35 mg/mL), with or without various concentrations of inhibitors/flavonoids, were prepared and kept on ice. The enzyme reaction was initiated by the addition of ≥20 mM sucrose, maltose or isomaltose and immediately incubating in a 37 °C water bath for 10 min (or various time points during method setup and validation). Following incubation, the enzyme activity was terminated by incubating in a 96 °C water bath for a further 10 min. A positive control without any added inhibitor/flavonoid was simultaneously tested in each batch, and negative controls without enzymes or substrates were also assayed to evaluate the stability of the inhibitors/compounds. Samples were prepared for HPAE-PAD analysis as described in Section 2.3. Specific enzyme activities were determined (U/mg CFE protein) and expressed as a percentage of control enzyme activities accordingly.

#### 2.6.1. Optimization of Assay Conditions and Enzyme Kinetics

Preliminary assays to optimize the substrate and enzyme concentrations and incubation time were performed to ensure enzyme kinetic experiments were carried out under initial linear velocity conditions (substrate depletion <10%). CFE protein concentration was tested, and specific activities found to be linear, at 0.10–0.35 mg/mL (Table 1), with substrates tested at 10–80 mM for maltase and 5–50 mM for sucrase and isomaltase, while incubation times were tested for 10–60 min. Michaelis-Menten and Lineweaver-Burk plots were used to obtain the kinetic parameters of the digestive enzymes, using GraphPad Prism 8.0 software (GraphPad Software, San Diego, CA, USA) (Table 2).

The specific activities of sucrase and maltase were similar to those we reported previously [27]. Lower CFE concentrations (0.25 mg/mL for sucrase and 0.1 mg/mL for maltase and isomaltase) were used to determine the substrate concentration required to achieve maximal catalytic efficiency or velocity of reaction (V_max_) (Table 2). Sucrase, maltase and isomaltase exhibited a linear production of glucose up to 40 min using 20 mM maltose, sucrose and isomaltose, respectively. A 10 min incubation time was used in the inhibition assays to ensure reactions were in the initial linear velocity phase. Assays were performed using CFEs from biological triplicates.

#### 2.6.2. Inhibition by Acarbose and Flavonoids

Using the optimal assay conditions, various concentrations of acarbose and flavonoids were tested. Controls (CFE and substrate only) were prepared by replacing the volume of inhibitor with SPB. Activities of sucrase, maltase and isomaltase were considered as 100% (or 0% inhibition) in the absence of an inhibitor. Compounds that exhibited enzyme inhibition of at least 15%, 25% and 50% were subjected to IC_15_, IC_25_ and IC_50_ value determination, respectively, or the maximum percentage of inhibition expressed. Estimation of inhibition values were determined in GraphPad Prism and the percentage of inhibition of the sample was calculated following Equation (7) below, where SA = specific activity:(7)$$\mathrm{Inhibition}\;\left(\%\right)=\frac{{\mathrm{SA}}_{\mathrm{control}\;}-{\;\mathrm{SA}}_{\mathrm{sample}}}{{\mathrm{SA}}_{\mathrm{control}}}\times 100$$

### 2.7. Data Analysis

Enzyme assays were performed at least once for CFEs from biological triplicates, with duplicate injections of each analyzed by HPAE-PAD. MS Excel (Microsoft, Redmond, WA, USA) was used for data processing and analysis. Final inhibition values were expressed as the percentage of control activity (%). The IC_15_, IC_25_ and IC_50_ values were calculated based on the plots created using GraphPad, using the dose-response inhibition (log (inhibitor) vs. normalized response—variable slope) model. The same software was used to determine the apparent K_m_ and V_max_ values under the enzyme–kinetic inhibition model, and for statistical analyses using non-parametric multiple comparisons tests. A difference was considered significant at *p* < 0.05 for all comparisons. All data are expressed as the mean ± SD or SEM, specified accordingly.

## 3. Results

### 3.1. Method Validation

The potential inhibition of key human intestinal α-glucosidases by flavonoids was evaluated. Initially, the analytical method was optimized and validated. Chromatograms for mixed sugar standards in H_2_O or in sodium phosphate assay buffer (SPB) were almost identical, with the same retention times and peak areas, demonstrating the absence of matrix effects (Figure 2a). Specificity was confirmed by comparing the retention times and peak areas of the sugars when run individually and as a mixture. The peak areas from mixed sugar standards prepared in SPB were used to plot standard curves (0–10 µg/mL), with excellent linearity for all sugars in this range (Figure 2b). The intercepts were not significantly different from zero (*p* = 0.225). LOD and LOQ for all five sugars were determined as a signal to noise ratio of 3:1 and 10:1, respectively. All sugar standards showed low LOD (ranging from 0.106 µM for maltose to 0.619 µM for fructose) and LOQ (ranging from 0.320 µM for maltose to 1.876 µM for fructose) (Table 3).

Intra- and inter-run precision was determined by analyzing the standards in triplicate in a single run on the same day and repeated on four different days within four months. Samples were kept at 4 °C at all times to avoid repeated freeze/thaw cycles or deterioration at RT. The mean peak areas and coefficients of variation (%CV) were calculated to determine the precision, as per ICH guidelines, [32] as presented in Table 4.

The intra-day precision range was calculated to be 0.63–2.14% for glucose, 0.94–5.52% for sucrose, 1.03–6.52% for fructose, 1.05–3.47% for isomaltose and 1.08–10.39% for maltose, while inter-day evaluations were 5.01–12.44%, 3.73–7.03%, 8.13–10.22%, 4.39–5.14% and 3.98–12.78% for glucose, sucrose, fructose, isomaltose and maltose, respectively. Precision results were considered excellent, as they fall below 15% of ICH guidelines, even at the lowest sugar concentration of 0.1 µg/mL. Extraction efficiency was between 95.9% and 108.9% for sucrose, maltose and isomaltose at two different concentrations, where the %RE_accuracy_ and the %CV_precision_ were <10% for all concentrations (Table 5).

Figure 2 displays the chromatogram of the sugar standards with efficient separation in a single 32-min elution, with good resolution. To analyze the efficiency of the enzyme reaction and sample extraction, substrates with or without cell-free extract (CFE) were digested, extracted and analyzed accordingly. The representative chromatograms in Figure 3 show the breakdown of sucrose into glucose and fructose (Figure 3a,b), and the breakdown of maltose (Figure 3c,d) and isomaltose (Figure 3d,e) into glucose was detectable with good resolution only when the relevant enzymes were present.

Table 6 indicates the peak areas of maltose, sucrose and isomaltose in the presence of (potential) inhibitors were not significantly different to when the compounds were not added (ANOVA: F < F critical one tail, *p* > 0.05), and without the presence of any additional glucose or fructose peaks in the chromatograms. High precision values were obtained for all tested sugars with inhibitors (%CV_precision_ < 15%), indicating no interference between sugars and tested substances. No peaks were observed in blank/control samples, where (potential) inhibitors, or assay buffer, or water, acetonitrile and eluent alone were tested.

### 3.2. Inhibition of α-Glucosidase Activities

Based on the evidence that some flavonols consumed in the diet may reach concentrations as high as 50 µM or more in the intestinal lumen [36], a concentration up to 200 µM, or maximal solubility, was used in this study. Where half-maximal inhibitory potential (IC_50_) could not be determined, IC_25_ and IC_15_ were calculated instead. All flavonols, acarbose and EGCG inhibited α-glucosidase activity to some extent (Figure 4, Figure 5 and Figure 6 and Table 7).

#### 3.2.1. Sucrase

On the basis of IC_25_ values, sucrase was inhibited in the order: acarbose > quercetagetin > galangin ≥ kaempferol > quercetin ≥ EGCG (Table 7), with >50% inhibition observed at the maximum tested concentrations of quercetin and quercetagetin (Figure 4).

#### 3.2.2. Maltase

Based on the IC_25_ values, the decreasing order of the maltase inhibitory activity of the studied inhibitors was acarbose ≥ quercetagetin ≥ galangin ≥ kaempferol ≥ EGCG > quercetin, where the significantly weakest inhibition of maltase was exhibited by quercetin (*p* < 0.05) (Table 7). Only acarbose and EGCG showed more than 50% maltase inhibition at their maximal tested concentrations (Figure 5).

#### 3.2.3. Isomaltase

On the basis of the IC_15_ values, the decreasing order of the isomaltase inhibitory activity of the studied inhibitors was concluded to be quercetagetin ≥ acarbose ≥ galangin ≥ kaempferol > quercetin ≥ EGCG. This pattern is almost identical to sucrase inhibition. The inhibition potentials of acarbose and EGCG were generally lower towards isomaltase than sucrase and maltase, where a few fold higher concentrations were required to exhibit half-maximal inhibition (Table 7). The inhibition shown by acarbose and EGCG was 96% and 71%, respectively, while all flavonols exhibited <30% inhibition of isomaltase (Figure 6).

## 4. Discussion

Flavonoids and other (poly)phenols potently inhibit α-amylase and α-glucosidase activities [37] without associated adverse gastrointestinal effects [38] and so may be useful in the management of T2D. However, previously, most inhibitory activities have been tested using α-glucosidase from yeast (*Saccharomyces cerevisiae*) with limited reports on enzymes of mammalian or human origin [37]. Multiple α-glucosidases are widely distributed in microorganisms, plants and animal tissues with variations in >20 amino acid sequences between species [39,40]. We explored disaccharide digestion in the human intestine by determining the inhibitory potential of flavonoids on sucrase, maltase and isomaltase in a specific mature Caco-2/TC7 clone, with high expression of SI [24]. Four flavonols were compared to a commercial α-glucosidase inhibitor, acarbose and a flavan-3-ol, EGCG. All compounds inhibited human sucrase, maltase and isomaltase in a dose-dependent manner.

### 4.1. Inhibitory Activities of Flavonoids

#### 4.1.1. Quercetagetin

Quercetagetin was first identified as part of spinacetin (quercetagetin-3’,6-dimethyl ether) in spinach [41], and a few recent reports identified a possible function in glucose metabolism [42,43,44,45]. Compared to quercetin, it has an additional C6-OH in the A ring, and exhibits various biological activities [46,47,48]. The additional C6-OH confers a strong affinity to proteins, speculated to weaken the binding of substrates to the active sites of enzymes and reduce or inhibit their activities [49]. Our study revealed quercetagetin as a strong human sucrase inhibitor, similar to acarbose and more potent than quercetin and EGCG. This was a greater inhibition than that seen previously against yeast α-glucosidase [42], demonstrating the varied activities between species and substrates used. Quercetagetin could be a promising α-glucosidase inhibitor, provided its high susceptibility to degradation [21] is considered.

#### 4.1.2. Kaempferol

Kaempferol has one less hydroxyl group in the B ring than quercetin. It has consistently shown lower inhibition than quercetin against rat maltase [50,51,52], rat sucrase [51,52] and porcine pancreatic α-amylase [50,52,53]. Inhibition of maltase by kaempferol (and galangin) was much less than that by acarbose in rats [54]. In contrast, kaempferol exhibited much stronger yeast α-glucosidase inhibition (95%) than acarbose [55,56,57,58], but weaker than quercetin [51,59]. In this study, kaempferol exhibited weaker inhibition of human sucrase, maltase and isomaltase than acarbose, with <32% inhibition at 40 µM, but was better than quercetin. A kaempferol-rich extract exhibited notably different IC_50_ values against human and yeast α-glucosidases [60]. These comparisons indicate again the varied inhibitory potentials between species.

#### 4.1.3. Galangin

Galangin has an unsubstituted B ring and is rich in many root plants, and possesses antiviral and anti-inflammatory properties with no toxic effects observed even at high doses in rats [61,62,63]. This compound regulates glucose homeostasis and enzymes responsible for glycolysis and gluconeogenesis in rats [64]. Reports have shown strong inhibition of α-glucosidase in yeast, better than acarbose [65] but similar to kaempferol [66]. In contrast, both galangin and kaempferol were poorer maltase inhibitors than acarbose when tested using enzymes from rat intestine [54]. Like kaempferol, the poor aqueous solubility of the aglycone is a drawback for practical use.

#### 4.1.4. Quercetin

Quercetin is a widely distributed flavonoid and therefore most researched in human and animal models. With a half-life of ~24 h [67], a few fold increase of this compound in plasma after several weeks of ingestion was noted (reviewed in [68]). Quercetin is a potent inhibitor of intestinal GLUT2, substantially reducing glucose absorption [69]. Quercetin has repeatedly been reported to inhibit yeast α-glucosidases more so than acarbose [42,50,70,71,72]. In contrast, for rat maltase and sucrase, it was shown to be weaker than acarbose (IC_50_ = 281.2 µM for maltase, IC_50_ > 400 µM for sucrase) [52], similar to the data reported here.

#### 4.1.5. EGCG

Among all tested flavonoids, while EGCG was most soluble with the highest inhibition reached for human sucrase (100%), maltase (68%) and isomaltase (71%) at 1500 µM, this is a supra-physiological concentration and EGCG generally exhibited the weakest inhibitory potential when compared by concentration alone, as indicated previously [30]. Conversely, much stronger α-glucosidase inhibitory effects were demonstrated against yeast or recombinant enzymes [73,74,75]. It has been suggested that EGCG (and quercetin) may exert much slower but more effective inhibition of disaccharide digestion in the intestine [75]. EGCG potently reduced glycaemic response in a diabetic animal model by binding to the active site of α-amylase and α-glucosidase [76] and decreased glucose uptake and GLUT2 expression in vitro [77]. However, a recent systematic review and meta-analysis from fourteen eligible articles demonstrated that the regular intake of EGCG-rich green tea had no significant effects on fasting blood glucose and insulin, HbA1c or HOMA-IR in T2D patients [78], which may be partially explained by the weak inhibition towards all three intestinal α-glucosidases shown in this study.

### 4.2. Structure-Function Relationships

The most active flavonol was quercetagetin, with IC_50_ values closest to acarbose. This suggests that stronger enzyme inhibition is observed with increasing hydroxyls on the A ring since quercetagetin is a stronger inhibitor than quercetin. Increasing hydroxylation of the B ring (from galangin to kaempferol to quercetin) improves the solubility of compounds but lowers inhibition. The lower aqueous solubility of flavonols, observed in quercetagetin, kaempferol and galangin, is a shortcoming of this study, and is the reason why some IC_50_ values could not be determined.

At the molecular level, the binding between hydroxyls in ring A, B or C of flavonoids to the active sites of α-glucosidases leads to structural changes in the enzyme evidenced by several docking studies with yeast α-glucosidase [20,55,72,79]. The inhibitory activity of flavonoids was concluded to be in the decreasing order of anthocyanidin ≥ isoflavone ≥ flavonol ≥ flavone ≥ flavonone ≥ flavan-3-ol [50], indicating the crucial role of A ring hydroxylation for potent α-glucosidases inhibition [80]. The A ring hydroxylation at C5 (fisetin converted to quercetin) or C6 (quercetin converted to quercetagetin) increased α-glucosidase and α-amylase inhibition in yeast and rat [50,81], and in human α-glucosidases as shown here.

The hydroxylation patterns, particularly 3-OH at a B ring catechol moiety, are among the major determinants of various biological effects of flavonoids [82]. The hydrophilicity of compounds is enhanced with increasing hydroxyls in the B ring, which also affects α-glucosidase inhibition, varying between species and substrates used [20,83]. The α-glucosidase inhibitory activity of flavonols increased with increasing hydroxyls on the B ring in rat and yeast (myricetin > quercetin > kaempferol) [50,51], in contrast to the results shown in this study (quercetagetin > galangin > kaempferol > quercetin > EGCG), and again emphasizing the importance of using human enzymes. Further, the hydrogenation of the C2 = C3 double bond in flavan-3-ols on the C ring weakened their enzyme inhibition activities [50,80], despite higher binding affinities [84]. Saturated C2-C3 bonds in flavan-3-ols are speculated to allow more twisting of the B ring and, together with additional hydroxyls on the gallate group in the C ring, increase solubility [85].

### 4.3. Comparing Flavonoids to Acarbose

Although mild α-amylase inhibition is beneficial for blunting glucose spikes, excessive inhibition may induce starch indigestion and abnormal bacterial fermentation, causing abdominal pain, bloating or cramping [22]. Acarbose can induce these undesirable effects due to its potent inhibition of human and mammalian pancreatic α-amylase [11]. Medicinal plant extracts containing quercetin and kaempferol consistently exhibited favorable inhibition against yeast and mammalian α-glucosidase over pancreatic α-amylase [53,86,87,88,89]. Many flavonoids have a higher inhibition of α-glucosidases, leading to a slow-release effect, than of α-amylase [75], which may be favoured over acarbose to decrease postprandial glucose spikes without the unpleasant side effects.

We have elucidated the inhibitory effects of flavonoids against human α-glucosidases compared with acarbose, influenced by structure, enzyme origin and substrates. A higher concentration of acarbose is required to inhibit maltase than sucrase, while all flavonoids showed similar inhibition of sucrase and maltase, in agreement with our previous findings using olive leaf extracts [30]. Isomaltose is known to be hydrolyzed slowly by the SI complex, reflected by the accumulation of isomaltose in the intestine [90]. We have shown that quercetagetin (at higher concentrations) inhibits starch digestion through direct α-amylase inhibition and starch complexation [91], making it a promising compound for regulating postprandial glycaemia.

Enzyme inhibition by plant extracts was consistently superior to acarbose when tested using yeast α-glucosidase [72,73], in contrast to data on human or mammalian sucrase and maltase. Previously, an IC_50_ of 20 µM for EGCG was determined for human maltase expressed in yeast [74], which is 8–10-fold lower than reported here, and by us previously [30], using human intestinal Caco-2/TC7 as the enzyme origin. This emphasizes the importance of using a relevant substrate and enzyme source when screening for inhibitory potentials of compounds.

## 5. Conclusions

Our study highlights the potential of selected flavonoids to inhibit human intestinal α-glucosidases, hence slowing carbohydrate digestion and reducing postprandial glycaemia. A sensitive and accurate method to determine sugar hydrolysis by sucrase, maltase and isomaltase has been successfully developed and validated. The use of HPAE-PAD to detect subtle changes in the concentrations of five sugars simultaneously, with minimal sample preparation and high precision within 32 min, has been central to this study. Acarbose and flavonoids exhibit different inhibition of human enzymes to those reported for yeast or mammalian α-glucosidases, emphasizing the need for a more pragmatic screening approach on individual human enzymes to elucidate their actual inhibitory potentials in vivo. Flavonoids from various sources are more effective against α-glucosidase than α-amylase [37]. The low solubility of some flavonoids limits the experimental concentration which can be employed, preventing the determination of IC_50_ values and necessitating the use of IC_25_ or IC_15_ values instead.

Quercetagetin, similar to acarbose, followed by kaempferol and galangin, exhibited greater inhibitory action against sucrase, maltase and isomaltase than EGCG and quercetin, although the latter compounds were more soluble in aqueous buffer. Two key structural elements of flavonoids for enhanced α-glucosidase inhibition in humans are the C6-OH A ring hydroxylation and reduced B ring hydroxylation. Improving understanding of how flavonoids bind to human α-glucosidases should provide a rational basis for exploiting antidiabetic compounds from dietary sources.

## Figures and Tables

**Figure 1 foods-10-01939-f001:**
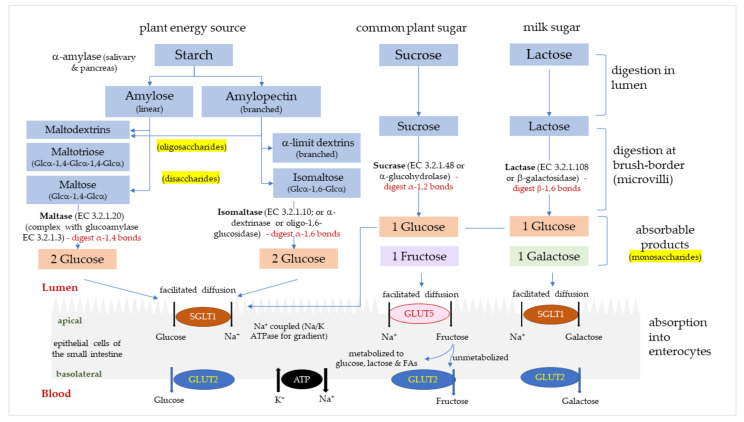
The digestion and absorption of starch and sugar in the small intestine. Glucose, fructose and galactose are absorbed into enterocytes via glucose transporters (GLUTs); sodium-glucose transport protein-1 (SGLT1) and glucose transporters -2 and -5 (GLUT2, GLUT5).

**Figure 2 foods-10-01939-f002:**
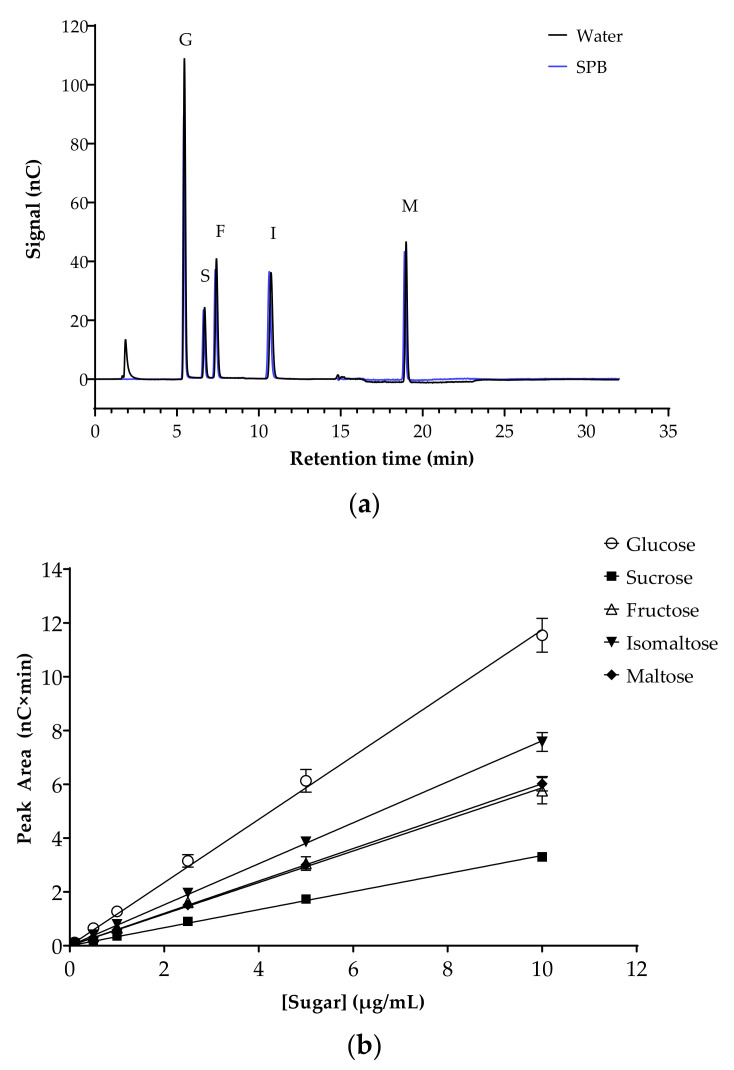
Separation of sugars, the substrates and products of the α-glucosidase assays, by HPAE-PAD ion chromatography. (**a**) Representative chromatogram of the mixed sugar standards in H_2_O (black line) and sodium phosphate buffer (SPB, blue line). Glucose (G), sucrose (S), fructose (F), isomaltose (I) and maltose (M) are at a concentration of 10 µg/mL each. Running conditions: 12 mM KOH eluent for 12 min, 100 mM KOH eluent for 8 min, 12 mM KOH eluent for 12 min (run time 32 min per injection); flow rate, 0.2 mL/min; injection volume 2.5 µL; column and compartment temperature, 30 °C and 20 °C, respectively. (**b**) Sugar standard curves (0–10 µg/mL). Data represent mean ± SEM of four replicates. All (R^2^) > 0.999. Error bars where not visible are smaller than the data point.

**Figure 3 foods-10-01939-f003:**
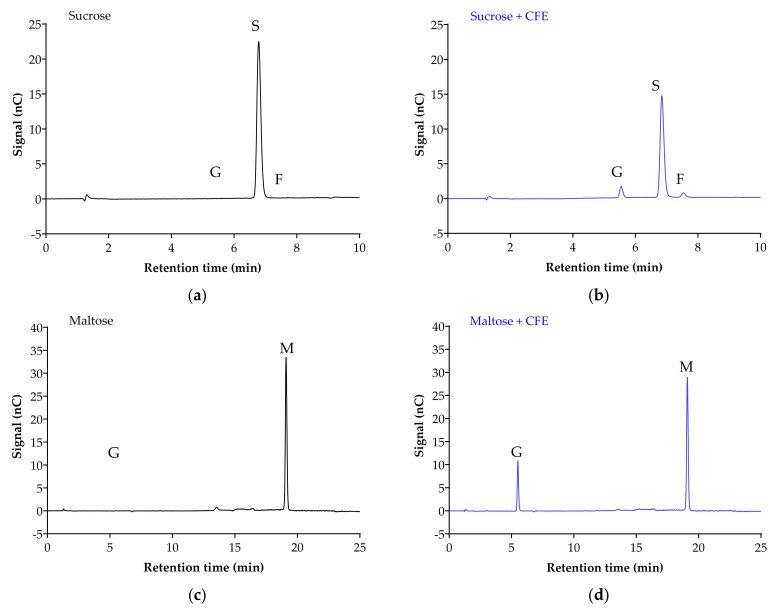
Representative chromatograms demonstrate the efficiency of sample digestion and extraction with or without α-glucosidases from Caco-2/TC7 cell-free extract (CFE). Sucrose breakdown into glucose + fructose (**a**,**b**), and maltose (**c**,**d**) and isomaltose (**e**,**f**) breakdown into glucose is evident in the presence of enzymes. Sample digestion and extraction conditions: substrate (20 mM) ± enzyme first incubated for enzyme reaction for 10 min at 37 °C, followed by second incubation for reaction termination for 10 min at 96 °C, then acetonitrile deproteination, sample dilution, sample injection and analysis by HPAE-PAD. Peaks were identified as glucose (G), sucrose (S), fructose (F), isomaltose (I) and maltose (M).

**Figure 4 foods-10-01939-f004:**
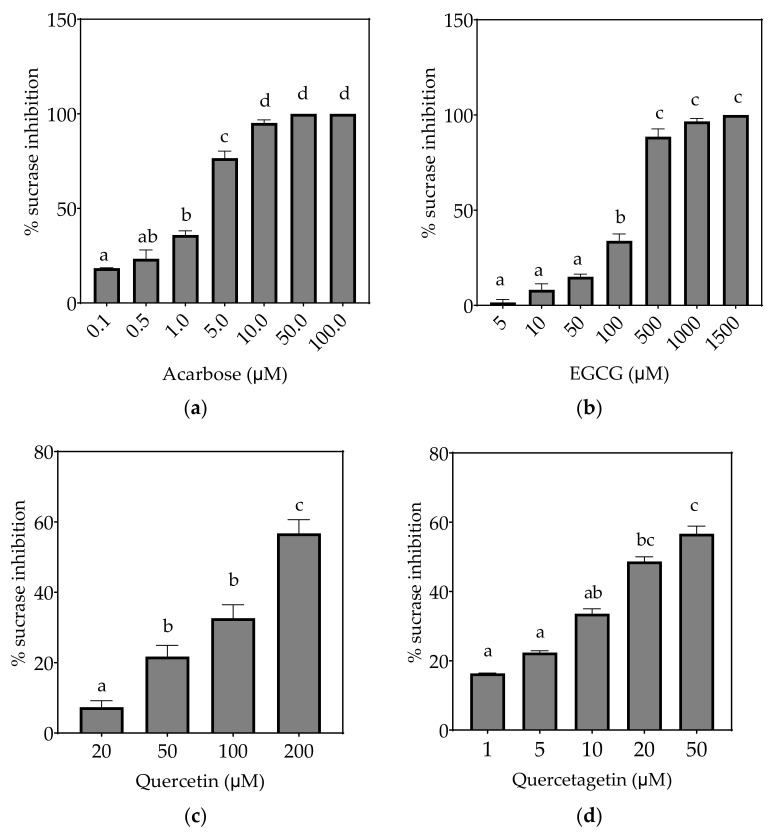
Inhibition of sucrase in Caco-2/TC7 cell-free extracts, using sucrose as the substrate, by (**a**) acarbose (positive control), (**b**) EGCG, (**c**) quercetin, (**d**) quercetagetin, (**e**) kaempferol and (**f**) galangin. Data are mean ± SEM, *n* ≥ 2 injections, for three different CFEs. Specific activity was calculated and relative inhibition was determined by comparing to the controls containing substrate and enzyme only. Values with different letters indicate significant differences (*p* < 0.05).

**Figure 5 foods-10-01939-f005:**
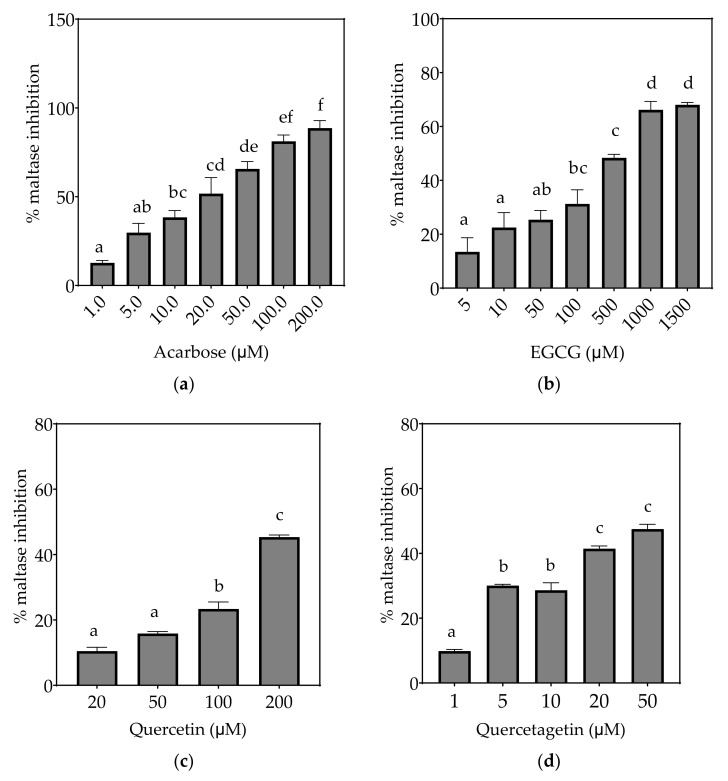
Inhibition of maltase in Caco-2/TC7 cell-free extracts, using maltose as the substrate, (**a**) acarbose (positive control), (**b**) EGCG, (**c**) quercetin, (**d**) quercetagetin, (**e**) kaempferol, and (**f**) galangin. Data are mean ± SEM, *n* ≥ 2 injections, for three different CFEs. Specific activity was calculated and relative inhibition was determined by comparing to the controls containing substrate and enzyme only. Values with different letters indicate significant differences (*p* < 0.05).

**Figure 6 foods-10-01939-f006:**
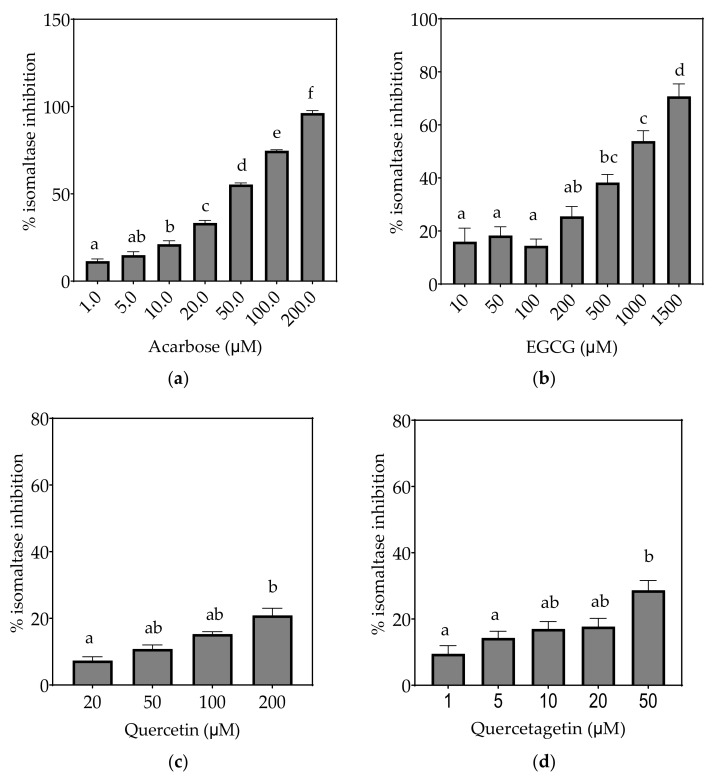
Inhibition of isomaltase in Caco-2/TC7 cell-free extracts, using isomaltose as the substrate, by (**a**) acarbose (positive control), (**b**) EGCG, (**c**) quercetin, (**d**) quercetagetin, (**e**) kaempferol and (**f**) galangin. Data are mean ± SEM, *n* ≥ 2 injections, for three different CFEs. Specific activity was calculated and relative inhibition was determined by comparing to the controls containing substrate and enzyme only. Values with different letters indicate significant differences (*p* < 0.05).

**Table 1 foods-10-01939-t001:** Linear regression showing the velocity of enzyme activities in Caco-2/TC7 cell-free extracts (CFE) (*n* = 3).

Parameters	Sucrase	Maltase	Isomaltase
Substrate concentration (mM)	20	20	20
CFE protein concentration (mg/mL)	0–0.35	0–0.25	0–0.30
Specific activity (mean mU/mg ± SD)	145 ± 31 ^1^	1197 ± 183 ^1^	298 ± 37
Linear equation	y=0.18x+0.001	y=0.94x+0.0002	y=0.63x+0.007
R^2^	0.9878	0.9995	0.9859

^1^ Reported previously as 130 ± 4 mU/mg or 725 ± 36 mU/mg using 10 mM sucrose or maltose, respectively, CFE ≤ 0.64 mg/mL [27].

**Table 2 foods-10-01939-t002:** Enzyme kinetic constants from Michaelis-Menten and Lineweaver-Burk plots for sucrase, maltase and isomaltase in Caco-2/TC7 cell-free extracts (CFE) using substrates sucrose, maltose and isomaltose, respectively (*n* = 3). App = apparent.

Parameters	Enzymes		
Sucrase (tested at one concentration)			
CFE tested (mg/mL)	0.25		
App K_m_ (mM) (95% CI)	5.8 (2.1–12.2)		
V_max_ (mmol/min) (95% CI)	0.039 (0.031–0.045)		
Maltase (tested at multiple concentrations)			
CFE tested (mg/mL)	0.10	0.20	0.30
App K_m_ (mM) (95% CI)	8.8 (6.4–11.7)	11.6 (8.8–14.9)	19.1 (13.9–26.1)
V_max_ (mmol/min) (95% CI)	0.35 (0.33–0.38)	0.64 (0.59–0.69)	1.04 (0.93–1.17)
Isomaltase (tested at multiple concentrations)			
CFE tested (mg/mL)	0.10	0.20	0.30
App K_m_ (mM) (95% CI)	3.1 (1.55–5.07)	8.8 (7.5–10.2)	9.3 (7.4–11.7)
V_max_ (mmol/min) (95% CI)	0.082 (0.076–0.087)	0.22 (0.21–0.22)	0.30 (0.29–0.32)

**Table 3 foods-10-01939-t003:** Calibration curve data for sugar standards (0.1–10 µg/mL), separated and analyzed by HPAE-PAD.

Sugar	Retention Time (min)	Retention Time Precision (%CV)	Calibration Range (μM)	Regression Equation	Correlation Coefficient (R^2^)	LOD ^1^ (μM)	LOQ ^1^ (μM)
Glucose	5.542	0.40	0.56–55.5	y = 1.1540x + 0.1366	0.9991	0.5952	1.804
Sucrose	6.842	0.60	0.29–29.2	y = 0.3299x + 0.0385	0.9994	0.4797	1.454
Fructose	7.808	1.05	0.56–55.5	y = 0.5741x + 0.8929	0.9990	0.6190	1.876
Isomaltose	10.975	0.50	0.29–29.2	y = 0.7562x + 0.0387	0.9999	0.2004	0.607
Maltose	19.492	0.93	0.29–29.2	y = 0.6027x + 0.0050	1.0000	0.1056	0.320

^1^ LOD (limit of detection) and LOQ (limit of quantification) were confirmed by injections of sugar standards in the range listed and measuring responses at 3 and 10 times the noise, respectively (LOD = 3.3 × (STEYX/SLOPE), and LOQ = 10×(STEYX/SLOPE)).

**Table 4 foods-10-01939-t004:** Intra- and inter-run peak area precision for sugars in mixed standard solutions analyzed by HPAE-PAD.

Sugar	Glucose		Sucrose		Fructose		Isomaltose		Maltose	
Concentration (µg/mL)	Peak Area (nC×min)	%CV	Peak Area (nC×min)	%CV	Peak Area (nC×min)	%CV	Peak Area (nC×min)	%CV	Peak Area (nC×min)	%CV
Intra-run (*n* = 3)—Repeatability										
0.1	0.13	2.14	0.04	5.52	0.07	6.52	0.08	3.47	0.05	10.39
0.5	0.69	1.47	0.20	2.73	0.36	4.18	0.42	2.19	0.31	3.10
1.0	1.38	1.21	0.39	1.56	0.70	2.20	0.84	1.63	0.62	2.17
2.5	3.40	1.75	0.95	1.98	1.72	1.67	2.04	1.66	1.60	2.46
5.0	6.59	1.36	1.82	1.43	3.27	1.63	3.99	1.48	3.16	1.35
10.0	12.20	0.63	3.42	0.94	6.06	1.03	7.79	1.05	6.27	1.08
Inter-day (*n* = 12)—Reproducibility										
0.1	0.12	12.44	0.04	7.03	0.06	10.22	0.08	5.14	0.06	12.78
0.5	0.66	6.66	0.19	5.83	0.35	9.37	0.41	5.10	0.30	7.00
1.0	1.29	7.55	0.37	4.34	0.67	7.69	0.80	4.84	0.60	5.40
2.5	3.19	6.59	0.91	4.30	1.64	8.26	1.97	4.33	1.53	5.39
5.0	6.17	6.14	1.74	3.60	3.06	7.99	3.85	4.15	3.03	4.71
10.0	11.58	5.01	3.32	3.73	5.76	8.13	7.56	4.39	6.05	3.98

Intra-run data collected from triplicate injections in a single run on 1 day; inter-run assays performed on four separate days in triplicate. The %CV of peak areas were <15%, indicating excellent precision and recoveries.

**Table 5 foods-10-01939-t005:** Extraction recoveries of maltose, sucrose and isomaltose added to assay buffer, analyzed by HPAE-PAD.

Criteria	Maltose		Sucrose		Isomaltose	
	C1	C2	C1	C2	C1	C2
Post-assay concentration recovered (mM)	10.89 ± 0.09	19.18 ± 0.23	10.06 ± 0.14	21.35 ± 0.30	10.82 ± 0.25	21.38 ± 0.28
Extraction recovery (%)	108.9 ± 0.9	95.9 ± 1.1	100.6 ± 1.4	106.9 ± 1.7	108.2 ± 2.5	106.9 ± 0.6
Relative error_accuracy_ (%RE_accuracy_)	8.93 ± 0.94	4.10 ± 1.14	1.96 ± 0.30	6.94 ± 1.68	8.24 ± 2.49	6.92 ± 0.62
Coefficient of variance_precision_ (%CV _precision_)	1.50	2.37	2.34	3.67	3.00	1.31

All values are mean ± SEM (*n* = 3). Pre-assay concentrations of maltose, sucrose and isomaltose spikes were 10 mM (C1) and 20 mM (C2). The %RE_accuracy_ and %CV _precision_ values are <10%, demonstrating excellent recoveries.

**Table 6 foods-10-01939-t006:** Comparison of peak areas of maltose, sucrose and isomaltose with and without acarbose or flavonoids.

Sugar (20 mM) ± Test Compounds	*n*	Peak Area ^1^ (nC×Time)	Precision (%CV)	One-Way ANOVA
Maltose only	9	5.37 ± 0.29	5.32%	
Maltose + Acarbose	6	4.91 ± 0.46	9.41%	
Maltose + EGCG	5	4.97 ± 0.24	4.73%	F (6,29) = 2.284
Maltose + Quercetin	4	5.02 ± 0.06	1.23%	F critical = 2.432
Maltose + Quercetagetin	4	5.15 ± 0.12	2.37%	*p*-value = 0.063
Maltose + Kaempferol	4	5.04 ± 0.09	1.78%	
Maltose + Galangin	4	5.36 ± 0.43	7.94%	
Sucrose only	10	2.97 ± 0.14	4.68%	
Sucrose + Acarbose	8	3.00 ± 0.18	5.83%	
Sucrose + EGCG	4	2.85 ± 0.27	9.53%	F (6,48) = 1.218
Sucrose + Quercetin	7	2.93 ± 0.16	5.37%	F critical = 2.295
Sucrose + Quercetagetin	3	2.97 ± 0.04	1.36%	*p*-value = 0.314
Sucrose + Kaempferol	5	3.09 ± 0.05	1.71%	
Sucrose + Galangin	7	3.01 ± 0.12	3.95%	
Isomaltose only	13	6.51 ± 0.70	10.73%	
Isomaltose + Acarbose	11	6.68 ± 0.67	10.11%	
Isomaltose + EGCG	10	6.36 ± 0.18	2.79%	F (6,48) = 1.218
Isomaltose + Quercetin	4	6.50 ± 0.45	6.86%	F critical = 2.295
Isomaltose + Quercetagetin	3	6.30 ± 0.02	0.32%	*p*-value = 0.314
Isomaltose + Kaempferol	7	6.96 ± 0.29	4.15%	
Isomaltose + Galangin	7	6.68 ± 0.49	7.29%	

^1^ Values are mean ± SD, *n* = technical replicates from biological CFE replicates of three. Acarbose and flavonoids were tested without enzymes at maximum concentrations: acarbose and quercetin, 200 μM; EGCG, 1500 μM; quercetagetin, 50 μM; kaempferol, 40 μM; galangin, 25 μM.

**Table 7 foods-10-01939-t007:** Structures and inhibitory concentrations of acarbose and flavonoids in the human Caco-2/TC7 intestinal cell model.

Compounds Tested	Drug	Flavan-3-ol	Flavonols
Acarbose	EGCG	Quercetin	Quercetagetin	Kaempferol	Galangin
Chemical structure	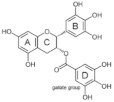	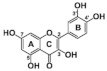	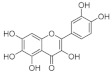	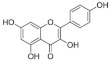	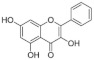
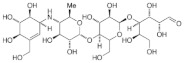
Molecular formula	C_25_H_43_NO_18_	C_22_H_18_O_11_	C_15_H_10_O_7_	C_15_H_10_O_8_	C_15_H_10_O_6_	C_15_H_10_O_5_
Concentration (µM)	0–200 ^1^	0–1500 ^1^	0–200 ^2^	0–50 ^2^	0–40 ^3^	0–25 ^3^
Ring position and substitution					
C3, C ring	-	galloyl	hydroxyl	hydroxyl	hydroxyl	hydroxyl
C5, A ring	-	OH	OH	OH	OH	OH
C6, A ring	-	H	H	OH	H	H
C7, A ring	-	OH	OH	OH	OH	OH
C3′, B ring	-	OH	OH	OH	H	H
C4′, B ring	-	OH	OH	OH	OH	H
C5′, B ring	-	OH	H	H	H	H
50% inhibitory concentration (IC_50_)					
Sucrase (µM)	1.65 ± 0.25 ^a^	175.2 ± 60.1 ^c^	161.9 ± 13.6 ^c^	21.7 ± 5.3 ^b^	ND (31%)	ND (33%)
Maltase (µM)	13.9 ± 2.3 ^a^	186.4 ± 40.4 ^b^	247.3 ± 7.0 ^b^	ND (48%)	ND (25%)	ND (22%)
Isomaltase (µM)	39.1 ± 2.1 ^a^	461.9 ± 60.3 ^b^	ND (18%)	ND (29%)	ND (27%)	ND (22%)
25% inhibitory concentration (IC_25_)					
Sucrase (µM)	0.60 ± 0.09 ^a^	72.9 ± 10.3 ^d^	69.5 ± 8.2 ^d^	6.6 ± 1.8 ^b^	30.3 ± 8.2 ^c^	20.8 ± 5.5 ^c^
Maltase (µM)	4.6 ± 0.8 ^a^	43.8 ± 9.2 ^b^	82.0 ± 4.6 ^c^	6.7 ± 1.5 ^a^	44.2 ± 4.4 ^b^	17.6 ± 2.2 ^a^
Isomaltase (µM)	14.4 ± 1.1 ^a^	241.9 ± 40.4 ^b^	ND (18%)	20.3 ± 7.3 ^a^	34.0 ± 8.4 ^a^	29.4 ± 1.6 ^a^
15% inhibitory concentration (IC_15_)					
Sucrase (µM)	0.32 ± 0.05 ^a^	41.2 ± 6.0 ^d^	40.5 ± 5.1 ^d^	3.5 ± 1.1 ^b^	14.2 ± 3.7 ^c^	12.7 ± 2.1 ^c^
Maltase (µM)	2.4 ± 0.4 ^a^	21.7 ± 4.5 ^c^	43.0 ± 3.1 ^d^	3.4 ± 0.9 ^ab^	19.0 ± 5.5 ^bc^	8.7 ± 1.1 ^abc^
Isomaltase (µM)	7.8 ± 0.7 ^a^	118.6 ± 18.0 ^b^	104.0 ± 18.8 ^b^	5.9 ± 1.9 ^a^	12.5 ± 3.7 ^a^	9.3 ± 1.2 ^a^

IC values (mean ± SEM, *n* = 3) with different superscript letters in the same row indicate significant differences (*p* < 0.05, nonparametric Kurskal-Wallis and Dunn’s multiple comparison tests). Compounds soluble either ^1^ SPB buffer, ^2^ DMSO or ^3^ ethanol. (ND) not determined, value in brackets indicates percent inhibition at maximum concentration tested. Flavonol skeletons show the numbering system of three rings A, B and C. Additional gallate group (D ring) is present in some flavan-3-ols.

## Data Availability

Not applicable.

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
