# Peer review of "Flavonoids as Human Intestinal α-Glucosidase Inhibitors"

_foods, 2021, doi:10.3390/foods10081939_

Round 1
Reviewer 1 Report
The inhibitory activities of four structurally-related flavonols (quercetin, kaempferol, quercetagetin and galangin) against sucrase, maltase and isomaltase in the extract of Caco-2/TC7 cells were evaluated by HPAE-PAD. The research was interesting and practical. The experiment was well organized. The results are supported by sufficient data to be convincing. But it needs some revisions:
- The source information of galangin and kaempferol should be provided.
- What is the effect of corresponding DMSO aqueous solution or ethanol solution for disolving flavonoids on enzyme activity?
- How about the contents and activities of sucrase, maltase and isomaltase in the cell-free extract ?
- How about the inhibition modes of these flavonoids for sucrase, maltase and isomaltase?
- Why the enzyme inhibition of EGCG with high aqueous solubility and many phenolic hydroxyl groups was the worst?
Author Response
- The source information of galangin and kaempferol should be provided.
Required information is added: see line #90-91 [Buffer components, sugar substrates and standards, and most inhibitors (acarbose, galangin, kaempferol and EGCG) were purchased from Sigma-Aldrich Corp., Merck]
- What is the effect of corresponding DMSO aqueous solution or ethanol solution for dissolving flavonoids on enzyme activity?
Required information is added: see line #140-144 [The maximum concentrations (v/v) of DMSO were ≤ 2% and ≤ 0.5% for quercetin and quercetagetin, respectively, and ethanol was ≤ 0.5% for kaempferol and galangin. The solvents did not affect enzyme activity, as demonstrated by vehicle controls.]
- How about the contents and activities of sucrase, maltase and isomaltase in the cell-free extract?
Required information is added: see line 259-262 [Using the optimal assay conditions, various concentrations of acarbose and flavonoids were tested. Controls (CFE and substrate only) were prepared by replacing the volume of inhibitor with SPB. Activities of sucrase, maltase and isomaltase were considered as 100% (or 0% inhibition) in the absence of an inhibitor.]
- How about the inhibition modes of these flavonoids for sucrase, maltase and isomaltase?
Based on the enzyme assay setup, whereby the inhibitors and enzymes (CFE) were combined initially and then the substrates were added to initiate the reactions, we cannot comment on the mode of inhibition and it was not the aim of the current paper but may form the basis of future studies.
- Why the enzyme inhibition of EGCG with high aqueous solubility and many phenolic hydroxyl groups was the worst?
This question has been addressed in line #487-489 and line #493-497, explaining the increased hydrophilicity with increased hydroxyl in B rings and saturation of C2-C3 bonds, therefore increased solubility but lower enzyme inhibition shown by EGCG.
Reviewer 2 Report
In the Manuscript ID: foods-1336080, the authors summarized that four structurally-related flavonols (quercetin, kaempferol, quercetagetin and galangin) were evaluated individually for their ability to inhibit human α-glucosidases (sucrase, maltase and isomaltase), and were compared with the antidiabetic drug acarbose and the flavan-3-ol (˗)-epigallocatechin-3-gallate (EGCG).
The paper is well-written, the tables and figures are of high quality, and the authors have clearly worked hard to produce a comprehensive dataset and detailed description of their methods.
As a limitation there is no indication of the mode of inhibition or of any suggestion of mechanism, also using fisetin and luteolin as flavonoids will help for SAR study; however they are fine for this paper which does not make any claims. But it needs some minor revisions:
#1 Style of some parentheses are not bold in the caption of Figure.
e.g. line297 (a), line301 (b)
#2 line376 a) should be (a)
#3 Caption in Figure 6. ... by a) acarbose (positive conTable 2. ... ? (b), (c)?
Author Response
#1 Style of some parentheses are not bold in the caption of Figure.
e.g. line297 (a), line301 (b)
#2 line376 a) should be (a)
#3 Caption in Figure 6. ... by a) acarbose (positive conTable 2. ... ? (b), (c)?
Minor changes have been made as suggested.